# The Management of Portal Vein Thrombosis after Adult Liver Transplantation: A Case Series and Review of the Literature

**DOI:** 10.3390/jcm11164909

**Published:** 2022-08-21

**Authors:** Liang-Shuo Hu, Zhen Zhao, Tao Li, Qin-Shan Li, Yi Lu, Bo Wang

**Affiliations:** Department of Hepatobiliary Surgery, First Affiliated Hospital of Xi’an Jiaotong University, No. 277, West Yanta Road, Xi’an 710061, China

**Keywords:** liver transplantation, portal vein thrombosis, case series

## Abstract

Background: Portal vein thrombosis (PVT) after adult liver transplantation (LT) is a rare but serious complication with no consensus on the ideal treatment. We report a case series and a comprehensive review of the literature on PVT after LT to discuss the therapeutic options. Methods: The clinical data of 360 adult patients (≥18 years of age) who underwent LT from January 2017 to January 2020 were reviewed, and a comprehensive search of PubMed and Web of Science was conducted. Patients diagnosed with PVT after LT were identified, and relevant risk factors and therapies were analyzed. Results: Among the 360 patients, 7 (1.94%) developed PVT after LT. Onset of PVT within one week after LT was found in six patients (85.71%). Four of the seven patients with PVT received systemic anticoagulation (low molecular weight heparin and warfarin) therapy. Minimally invasive interventional therapies combined with systemic anticoagulation (heparin and warfarin) were applied for three patients, two of whom died because of severe abdominal hemorrhage and liver failure. Of the 33 cases reported in the literature, minimally invasive interventional therapy combined with systematic anticoagulation or sclerotherapy were the most-used methods (20/33). Systemic anticoagulation was administered to four patients, and surgical operation (thrombectomy; portosystemic shunt and retransplantation) was performed for nine patients. Among these 33 patients, 4 eventually died. Conclusions: Interventional therapy combined with systemic anticoagulation is a good choice for the management of PVT after LT, and in our experience, systemic anticoagulation alone can also have a positive effect for early PVT patients.

## 1. Background

The establishment of uncompromised inflow and adequate venous outflow are critical for successful liver transplantation (LT). However, the prognosis after LT can be marred by serious vascular complications, including hepatic artery thrombosis (HAT) and portal vein thrombosis (PVT), which can significantly increase posttransplant mortality [1,2]. PVT can be more detrimental to the graft and patient survival than HAT after LT because an alternative portal inflow is difficult to establish [3]. PVT is a rare complication with a reported incidence of 3% to 7% in patients who have undergone LT, and it is somewhat more frequent in children [4,5]. A diagnosis of PVT after LT usually depends on follow-up ultrasonography and the manifestation of hepatic dysfunction or portal hypertension involving variceal hemorrhage, splenomegaly, and hypersplenism [6,7,8,9]. Several predisposing factors are considered relevant to post-LT PVT formation, including preoperative PVT in the recipient, previous portosystemic shunts or splenectomy, hypercoagulable state, technical issues during the LT, vein graft and conduit availability, and pediatric transplantation [10,11,12,13]. 

Management of PVT aims to relieve the presenting symptoms and maintain normal liver function. Reported therapeutic options for PVT include systemic anticoagulation, management of portal hypertension with banding and sclerosis of bleeding varices, portosystemic shunting, minimally invasive interventional therapies, surgical thrombectomy, and retransplantation [14,15]. However, with only a few articles available to guide medical options, the ideal treatment of PVT after LT has not been defined. Here, we report a case series and a comprehensive review of the literature on PVT after LT to discuss the therapeutic options. Incidence, risk factors, clinical features, and treatment outcomes are described.

## 2. Methods

### 2.1. Clinical Case Series

The medical records of all adult patients (18 years of age or older) who received donor organs after cardiac death (DCD) and who underwent LT at the First Affiliated Hospital, Xi’an Jiaotong University from January 2017 to January 2020 were reviewed. The clinical data of these recipients were collected, including demographic features, perioperative laboratory values, postoperative complications, and PVT management after LT. Patients diagnosed with PVT after LT were identified, and relevant risk factors were analyzed, including surgical history and the presence of PVT before LT. Written consent had been given by the patients for their information to be stored in the hospital database and used for research. This study was approved by the First Affiliated Hospital of Xi’an Jiaotong University Ethics Committee. 

### 2.2. Review of the Literature on PVT after LT

We performed a comprehensive search of PubMed and Web of Science from their inception to 31 January 2022 to identify all available articles that discussed the management of PVT after LT in adult patients. We combined search keywords “portal vein thrombosis” (or “PVT”), “vascular complication”, “adult”, and “liver transplantation” (or “LT”). All case reports and case series discussing PVT after LT in adults were included. A secondary search of the bibliography of each included article was also undertaken. Excluded studies were those with insufficient data. Abstracts, review articles, editorials, and letters were also excluded. Relevant data including patient demographics, clinical symptoms, management, and outcomes were extracted from the included publications.

### 2.3. Statistical Analysis

Continuous data with normal distributions were reported as mean ± standard deviations (SD). Abnormal distribution variables were expressed as medians and ranges. Categorical data were reported as frequencies and percentages. All statistical analyses were performed using IBM SPSS (version 20.0, IBM, Armonk, NY, USA).

## 3. Results

### 3.1. Clinical Case Series 

There were 360 adult patients who underwent LT from 1 January 2017 to 1 January 2020 at the First Affiliated Hospital of Xi’an Jiaotong University. Among these patients, seven (1.94%) developed PVT after LT, with a male-to-female ratio of 2:5. The mean age was 54.74 ± 9.68 years (median: 50.0; range: 44.0–57.0). The etiology included HBV-related liver cirrhosis in four patients, hepatic carcinoma post-HBV infection in one patient, and autoimmune liver disease in two patients. 

Of these seven patients, the onset of PVT formation occurred within one week post transplantation in six patients (85.71%). Two of these six patients experienced liver function injury with elevated transaminases and bilirubin; the remaining four patients were asymptomatic, with diagnosis confirmed by ultrasonography. PVT was found 3 months after LT in one patient: a 57-year-old woman who was readmitted with upper gastrointestinal tract bleeding. Two patients (28.57%) had undergone a splenectomy before LT. Pretransplantation PVT was found in six patients (Yerdel stage I: 3; stage II: 1; and stage III: 2) [16] (For Yerdel stage: Grade I: <50% of light, with no or minimal obstruction of the superior mesenteric vein; Grade II: Grade I with obstruction > 50%, including total obstruction; Grade III: Complete obstruction of the portal vein and proximal superior mesenteric vein; and Grade IV: Complete obstruction of the portal vein and superior mesenteric vein).

Four of these seven patients received systemic anticoagulation in the form of low molecular weight heparin and warfarin therapy. The median follow-up duration was 17.6 months (range: 5.7–36.4 months). Minimally invasive interventional therapy combined with heparin and warfarin systemic anticoagulation was applied for the three remaining patients. Two of these three patients died because of severe abdominal hemorrhage and liver failure. A summary of the case series is provided in Table 1.

### 3.2. Review of the Literature on PVT after LT

A total of 25 articles reported one or more cases, for a total of 33 cases of post-LT PVT formation in adult LT recipients [2,3,10,11,12,17,18,19,20,21,22,23,24,25,26,27,28,29,30,31,32,33,34,35,36]. The characteristics of the reported cases are summarized in Table 2. Among these 33 cases, the etiologies for LT were liver malignancies in 6 cases (18.18%), primary sclerosing cholangitis in 5 cases (15.15%), post-infection liver cirrhosis in 7 patients (21.21%), alcohol-related cirrhosis in 8 patients (24.24%), and other non-neoplastic liver disease in 7 patients (21.21%, Table 2). The median age was 53.0 years (range: 23.0–72.0 years), and 60.60% were male. The time to PVT onset from LT ranged from 1 day to 10 years, and the median follow-up was 10 months (range: 1.0–120 months) (Table 3). Among these 33 patients, splenectomy was performed in 5 patients, transjugular intrahepatic portosystemic shunt (TIP) or splenorenal shunts were placed in 3 patients, coronary vein steals were reported in 3 patients, and transarterial chemoembolization was reported in 3 patients. Data on pretransplantation PVT status was reported in 10 of the 33 cases, 4 of which (40%) reported pretransplantation PVT. In total, 15 (45.45%) of the 33 patients had relevant risk factors that may have predisposed them to the formation of PVT after LT (Table 2).

Of the reported 33 cases, 21 (63.63%) were diagnosed within 30 days post-LT, and ultrasonography was the first choice for monitoring PVT formation after LT. The symptoms of these cases for evaluation are shown in Figure 1. Injury to liver function was the most frequent manifestation (15/33), and 6 of 33 cases were asymptomatic. Regarding the therapeutic choices for PVT after LT, minimally invasive interventional therapy combined with systematic anticoagulation or sclerotherapy were the most frequently used methods (20/33; Table 3). Among these 20 patients, 4 eventually died. Two patients died of liver-related complications (one due to bleeding and one due to multiple organ failure). The remaining two patients died of rejection after discontinuing immunosuppression drugs and a fungal brain abscess. In the other 13 patients, systemic anticoagulation was applied for 4, and surgical operation (thrombectomy, portosystemic shunt, and retransplantation) was performed in 9 patients.

## 4. Discussion

Portal vein thrombosis following adult liver transplantation is an uncommon complication. Preoperative recipient portal vein thrombosis is considered a primary risk factor predisposing recipients to PVT formation after LT (18). Shaked and Busuttil [37] found that 2 of 33 preoperative PVT patients developed PVT following LT, and Davidson et al. [38] reported that the incidence of rethrombosis is approximately 21% in patients with preoperative PVT. These findings are in accordance with our results. Among the 360 LT patients at our institution, 51 (14.2%) had PVT before LT. The incidence of rethrombosis was 9.8% (5/51), which is much higher than the incidence in patients without pre-LT PVT (0.65%). In our current case series, six of seven patients who developed PVT following LT had preoperative PVT. The study by Shaked and Busuttil also reported that the incidence of PVT was highest among patients with pre-existing pathologies of the portal vein, including chronic active hepatitis, hypercoagulable states, trauma or previous dissection of the porta hepatis, and splenectomy (37). Other factors include technical problems such as misalignment or excessive vessel length, ongoing rejection, liver fragment transplantation, using venous conduits in PVT, portosystemic shunts, and decreased portal flow from stenosis [5,39,40,41,42,43,44,45]. In our institution, 2 of 20 (10%) patients who had undergone a splenectomy before LT developed PVT after LT. Of the remaining 340 patients without splenectomy before LT, only 5 (1.47%) developed post-LT PVT. In terms of reported cases, 15 (45.45%) of the 33 patients had relevant risk factors, including preoperative PVT (4/33), splenectomy (5/33), portosystemic shunts (3/33), coronary vein steal (3/33), and living-donor liver transplantation (4/33).

PVT can occur early or late after LT. Early PVT was defined as PVT detected within 30 days of LT. Late PVT was defined as PVT detected more than 30 days after LT or as early PVT persisting after 30 days [46]. In our current case series, 85.71% (6/7) of patients were diagnosed with early PVT. Among the cases included in the literature review, 63.64% were confirmed within 30 days. The clinical manifestation may differ between early and late phase PVT. PVT formation early after LT may present as mild or severe graft dysfunction, variceal bleeding, encephalopathy, and, rarely, bowel infarction requiring immediate intervention. Liver function injury was the most frequent symptom in our study. Bakthavatsalam et al. reported that early PVT formation was associated with high morbidity and mortality following LT [19]. Patients with late-phase PVT after LT may present with liver dysfunction or portal hypertension, or they may be asymptomatic due to the development of collateral circulation that reestablished portal flow to the liver [47]. Collateral circulation developed in one patient in our study and played an important role in the patient’s recovery of graft function. Because several patients were asymptomatic, ultrasonography played an important role in the diagnosis of postoperative PVT. For patients with risk factors, Doppler ultrasonography should be used to monitor portal blood flow once a day in the first week, once every 2 days in the second week, once every 2 weeks after 1 month, once every month after 3 months, and once every 3 months after 1 year. For patients at high risk of thrombosis, it is recommended to review the abdomen with contrast-enhanced CT in 3 to 6 months.

The treatment strategies for PVT after LT should depend on clinical manifestations and laboratory tests. Reported therapies range from systemic anticoagulation with heparin to transjugular or percutaneous transhepatic catheter-based thrombolysis to portosystemic shunts, surgical thrombectomy, and retransplantation [31,48,49]. However, optimal treatment guidelines for posttransplantation PVT have yet to be defined. Currently, interventional radiological procedures are becoming attractive alternatives to surgical thrombectomy or retransplantation for the management of PVT following LT due to their minimal invasiveness, low rates of complications, and high success rates. There are a number of different interventions described in recent years from studies on portal vein obstruction after pediatric liver transplantation, including percutaneous transluminal angioplasty (PTA), stent placement, mesorex bypass (MRB), and endovascular recanalization (EVR) [50]. In the reviewed literature, interventional radiology therapy combined with systematic anticoagulation (48.49%) was the most frequently chosen method for management of PVT, and included PTA, stent placement, and transjugular intrahepatic portosystemic therapy. Transluminal angioplasty combined with thrombolysis and stent placement is the mainstream treatment for those with PVT in the acute period. However, the technical difficulty of percutaneous access or cannulation of a portal vein with thrombosis limits the application of interventional therapy. It has also been reported that potential anastomotic disruption is a risk for the application of interventional procedures in the early posttransplant period [3]. Regarding anticoagulation, LMWH is the initial treatment of choice, and vitamin K antagonists (VKAs) can be used for long-term treatment. In some retrospective analyses, DOACs showed better efficacy and safety than traditional anticoagulants such as warfarin against PVT in cirrhotic patients [6]. Currently, there is no evidence for DOACs in the management of PVT after adult LT. We think DOACs are a good prospect, but well-designed randomized controlled trials are still needed to further evaluate their safety and efficacy in this type of patient. Currently, there is still a lack of data on the treatment course of anticoagulation for patients with PVT post-LT. We can only consider the treatment strategy for PVT in patients with cirrhosis. Guidelines suggest a treatment course of 6 months without contraindications [1]. In order to prevent re-thrombosis, particularly in patients with risk factors such as superior mesenteric vein thrombosis or with a past history suggestive of intestinal ischemia, lifelong anticoagulation is recommended [2]. For patients who have undergone TIPS or stent placement, anticoagulation therapy is controversial because the interventional treatment alone may have a good recanalization rate [3]; six months of anticoagulation is still recommended to prevent re-thrombosis for these patients. However, the clinical evidence for these problems is inadequate, and data from more clinical trials are needed to identify the best treatment for patients with PVT post-LT.

In our experience, systematic anticoagulation should be used immediately for PVT patients with stable liver function. Systematic anticoagulation alone with careful monitoring is a good option for PVT patients early after liver transplantation. In addition, interventional radiology is also a good choice if the thrombus remains unchanged after anticoagulation. However, manipulation of percutaneous access or cannulation of the portal vein should be performed carefully to avoid anastomotic disruption and bleeding. In the presented cases, four patients received anticoagulation treatment and recovered uneventfully. One patient died of uncontrolled bleeding following transjugular transhepatic portal vein puncture thrombolysis. For patients with PVT formation more than 30 days posttransplantation, anticoagulation and interventional treatment can also be chosen for thrombolysis if collateral circulation has already formed and symptoms are mild.

There are several limitations to our study due to the limited number of cases. Because of the characteristics of case reports or series, the risk factors of posttransplant PVT formation cannot be fully analyzed. In addition, owing to the selection biases of retrospective cases, mortality may be underestimated.

## 5. Conclusions

In summary, the management of PVT following LT should depend on the clinical symptoms and the patient’s condition. Systemic anticoagulation alone can have a positive effect on early PVT patients from our experience. Interventional therapy combined with systemic anticoagulation is reported to be a good choice, but anastomotic disruption and bleeding should be carefully avoided. For the effects of conservative treatment and surgery, such as thrombectomy, portosystemic shunts, or retransplantation, further prospective studies with larger sample sizes are needed.

## Figures and Tables

**Figure 1 jcm-11-04909-f001:**
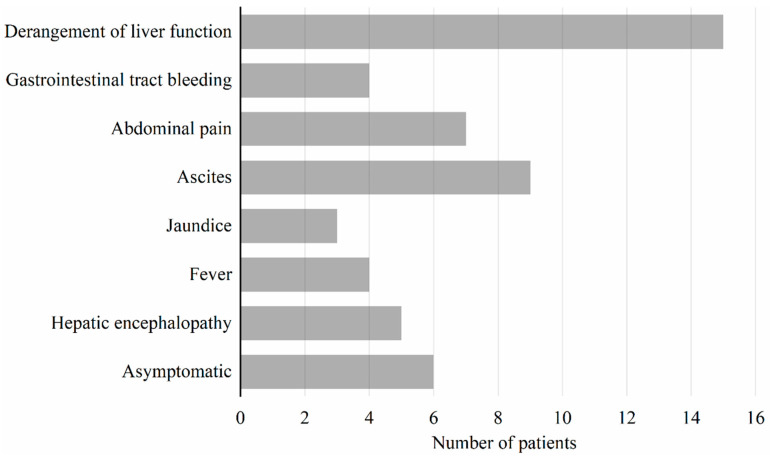
Symptoms present in patients with portal vein thrombosis after liver transplantation. For the 33 patients with portal vein thrombosis after liver transplantation reviewed in the literature, most-to-least common symptoms were derangement of liver function (15/33, 45.5%), ascites (9/33, 27.3%), abdominal pain (7/33, 21.2%), hepatic encephalopathy (5/33, 15.2%), gastrointestinal tract bleeding (4/33, 12.1%), fever (4/33, 12.1%), and jaundice (3/33, 9.15); however, 6 patients (6/33, 18.2%) were asymptomatic.

**Table 1 jcm-11-04909-t001:** Characteristics of patients with PVT after orthotopic liver transplantation.

Patient	Age (Years)	Gender	Etiology	MELD Score	Pre-LT PVT (Yerdel Stage)	Surgery History	Other Risk Factors	Type of LT	Post-LT PVT		PVT Onset (Days)	Treatment	Prognosis
									Position	Yerdel Stage			
1	52	Female	HBV-Liver Cirrhosis; HCC	30	yes/III	TACE	No	OLT	Main PV; SMV	III	2	Anticoagulation	Alive
2	44	Female	HBV-Liver Cirrhosis	33	yes/II	No	No	OLT	Main PV	II	7	Anticoagulation	Alive
3	44	Female	HBV-Liver Cirrhosis	23	yes/I	Total Hysterectomy	No	OLT	Main PV	II	1	Anticoagulation	Alive
4	50	Female	Autoimmune Liver Disease	16	yes/I	Splenectomy	No	OLT	Main PV; SMV	III	1	Percutaneous transhepatic angioplasty thrombolysis+ anticoagulation	Alive
5	57	Female	Autoimmune Liver Disease	22	0	No	No	OLT	Main PV	II	90	Anticoagulation+ percutaneous transhepaticportal venous thrombolytic therapy	Dead
6	55	Male	HBV-Liver Cirrhosis	11	yes/III	No	No	OLT	Main PV; SMV	II	1	Transjugular transhepatic portal vein puncture thrombolytic+ anticoagulation	Dead
7	46	Male	HBV-Liver Cirrhosis	12	yes/I	Splenectomy	No	OLT	Main PV; SMV	I	1	Anticoagulation	Alive

Abbreviations: HBV, hepatitis B virus; HCC, hepatocellular carcinoma; MELD, model for end-stage liver disease; OLT, orthotopic liver transplantation; PV, portal vein; SMV, superior mesenteric vein; TACE, transcatheter arterial chemoembolization.

**Table 2 jcm-11-04909-t002:** Characteristics of studies included in this review.

Author	Country	Patients	Etiology of LT	Type of LT	PVT Diagnosis	Pre-LT PVT	Surgery History	Other risk Factors
George et al. (1988) [12]	USA	1	Primary sclerosing cholangitis	OLT	US	NP	Proximal choledochoduodenostomy	NP
Khan et al. (2014) [2]	USA	2	Cholangiocarcinoma; Alcohol-related cirrhosis	OLT + PBLT	US+CT+MRI	NP	Pancreatoduodenectomy (one patient)	Hypercoagulable
Gill et al. (2009) [17]	UK	1	Cryptogenic cirrhosis	OLT	US+CT	Yes	Splenectomy	NP
Koo et al. (2008) [18]	Korea	1	Alcohol-related cirrhosis	OLT	US+CT	No	NP	Coronary vein steal
Kensinger et al. (2014) [3]	USA	1	Reformed ethanol abuse	OLT	US+CT	NP	TIPs	NP
Cherukuri et al. (1998) [10]	USA	2	HCV+Primary sclerosing cholangitis; Postinfectious cirrhosis	OLT + ReLT	US	NP	Splenorenal shunt (one patient); LT	NP
Bakthavatsalam et al. (2001) [19]	USA	1	Autoimmune hepatitis	OLT	US	Yes	NP	NP
Barriga et al. (2004) [20]	Italy	1	HBV liver cirrhosis	PBLT	US	No	Splenectomy	NP
Guckelberger et al. (1999) [21]	Germany	1	HBV liver cirrhosis	OLT	US+MRI	No	Cholecystectorny	NP
Brown et al. (2013) [22]	USA	1	HCV+Alcohol abuse	PBLT	US+CT	NP	Splenectomy	NP
Lodhia et al. (2010) [23]	USA	3	Alcohol-related cirrhosis; Cryptogenic cirrhosis; Primary sclerosing cholangitis	OLT	MRI	NP	Proctocolectomy (one patient)	NP
Haska et al. (1993) [24]	USA	1	Postinfectious cirrhosis	ReLT	US	No	LT	Coronary vein steal
Kawano et al. (2016) [25]	Japan	1	HCC	Living-donor LT	CT	No	Splenectomy	NP
Jeng et al. (2014) [26]	China	2	HCC; Alcohol-related cirrhosis	Living-donor LT	CT	NP	NP	NP
Kobayashi et al. (2012) [27]	Japan	1	Autoimmune hepatitis	Living-donor LT	Endoscopy	NP	Splenectomy	NP
Durham et al. (1994) [11]	USA	3	Alcohol-related cirrhosis; Drug abuse+postnecrotic cirrhosis; Autoimmune hepatitis	OLT	US+Arterial portography	NP	Distal splenorenal shunt (one pantient)	NP
Daniel et al. (1997) [28]	USA	1	Primary sclerosing cholangitis	OLT	US+CT	NP	NP	NP
Eric et al. (1990) [29]	USA	2	Cholangiocarcinoma; Primary sclerosing cholangitis	OLT + ReLT	CT+ SMA portography	NP	Choledochojejunostomy; LT	NP
Bhattacharjya et al. (1999) [30]	UK	1	Cryptogenic cirrhosis	PBLT	US	Yes	NP	NP
Baccarani et al. (2001) [32]	Italy	1	HCV cirrhosis	PBLT	US	No	NP	NP
Ciccarelli et al. (2001) [31]	Belgium	1	HCV cirrhosis	OLT	US	Yes	NP	NP
Hung et al. (2020) [33]	China	1	NP	Living-donor LT	US+CT	NP	NP	NP
Centonze et al. (2020) [35]	Italy	1	HCC+Alcohol-related cirrhosis	OLT	CT	NP	NP	Coronary vein steal
Sribenjalux et al. (2019) [34]	Thailand	1	HBV liver cirrhosis+HCC	OLT	CT	NP	TACE	NP
Dumortier et al. (2019) [36]	France	1	Alcohol-related cirrhosis	OLT	CT	NP	NP	NP

Abbreviations: LT, liver transplantation; OLT, orthotopic liver transplantation; ReLT, retransplantation; PBLT, piggyback liver transplantation; HCV, hepatitis C virus; HCC, hepatocellular carcinoma; US, ultrasonography; CT, computed tomography; MRI: magnetic resonance imaging; SMA, superior mesenteric artery; PVT, portal vein thrombosis; TACE, transarterial chemoembolization; NP, not reported.

**Table 3 jcm-11-04909-t003:** Characteristics and treatment of reported cases of PVT after adult liver transplantation.

Variables	PVT after LT (n = 33)
Age (years)	53 (range: 23–72)
Gender (male/female)	20 (60.6%)
PVT position	
Main PV	12 (36.4%)
Partial PV	4 (12.1%)
PV + SMV	4 (12.1%)
PV + SMV + Splenic vein	7 (21.2%)
Onset time post-transplant	
≤30days	21 (63.6%)
>30days	12 (36.4%)
Treatment	
Anticoagulation/Non-treatment	4 (12.1%)
Anticoagulation+Interventional therapy	16 (48.5%)
Interventional therapy+Sclerotherapy	4 (12.1%)
Anticoagulation+Surgical thrombectomy	1 (3.0%)
Surgical thrombectomy+Interventional therapy	5 (15.2%)
Portosystemic shunt	1 (3.0%)
Retransplantation	2 (6.1%)
Follow-up duration (months)	10 (1–120)
Mortality	4 (12.1%)

Abbreviations: LT, liver transplantation; PVT, portal vein thrombosis; PV, portal vein; SMV, superior mesenteric vein.

## Data Availability

The study did not report any data.

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
