# Peer review of "The Management of Portal Vein Thrombosis after Adult Liver Transplantation: A Case Series and Review of the Literature"

_jcm, 2022, doi:10.3390/jcm11164909_

Round 1
Reviewer 1 Report
In this manuscript, Liang shuo Hu et al describe the optimal management of portal vein thrombosis after liver transplantation in adults, through a case series and a review of the literature.
I think that although it is a very specific issue, it is relevant since it is a serious complication that can compromise the viability of liver transplantation and there is little evidence of the most appropriate approach. The manuscript is well written and important concepts are especially clear. I think your post should be considered, depending on the responses to the following comments.
General comments
- At various points in the text, the use of minimally invasive interventional therapies is described as a treatment possibility. However, these therapies are not described or explained. I would be grateful if you could explain yourself in order to better understand the text.
- An important point would be to know how long we should maintain anticoagulation in this type of patient, however, there is no information on this point. I would appreciate it if you would review this point and make the necessary changes to the text.
- Nothing is mentioned in the article about the possibility of using DOACs in this group of patients. What evidence exists on the use of DOACs in this type of patient?
Specific comments
- The title is missing in the text. I would appreciate it if you included it.
- In relation to treatment, in the series of clinical cases, it is described that 3 of the 7 received thrombolysis and anticoagulation, and it is surprising that 2 of these died. I would appreciate reviewing this point, and explaining why you believe everyone in the intervention and anticoagulation group dies.
- In the discussion, it is described that if the hepatic artery does not present stenosis or thrombosis, anticoagulation is a good option. However, it is not previously explained why the hepatic artery plays this role. I would appreciate an answer and make the pertinent changes in the exrto
- The conclusions highlight that interventional treatment combined with anticoagulation is ALWAYS a good option. However, the results in relation to mortality from bleeding are not very good. I would like you to justify this point and make the appropriate corrections in the text.
Yours faithfully,
Reviewer 2 Report
The authors nicely outline a case series and review of the literature on portal vein thrombosis after liver transplantation. In addition, they discuss therapeutic options.
A couple of points to address:
- Background: Reported therapeutic options for PVT included “management of portal hypertension with sclerosis of bleeding varices and portosystemic shunt” – what about variceal banding?
- It was stated that “Two of these six patients experienced a liver function injury” – please define what you mean by liver function injury
- Consider commenting on if you believe there would be a benefit to a screening protocol for ultrasound with doppler post-transplant given some patients are asymptomatic.
- Comment on if any patients were started on a direct-acting oral anticoagulants
- Any data on the timing of anticoagulant use in relation to interventional radiology thrombectomies?
- For table 1: please include data on the type of liver transplant, any pre-LT PVT and other risk factors (similar to table 2).
Minor points to address:
- You mention benign diseases that cause liver - What are benign liver diseases that cause liver transplantation?
- Please use the less stigmatizing term ‘alcohol-related cirrhosis’ instead of ‘alcoholic cirrhosis’ throughout
- Please define all abbreviations prior to use (e.g., TIPS)
- In the review of the literature – add an s to make it patients and use was instead of were - “transarterial chemoembolization were reported in three patient”
- This sentence is missing a space between patients and died: “Two patientsdied of liver-related complications (one due to bleeding and one due to multiple organ failure).”
- This sentence at the end is non-specific: “All patients were in good condition during the follow-up duration.” – please either better define what this means or exclude it.
- This sentence is confusing: “Patients with late phase PVT after LT may present with liver dysfunction, or portal hypertension, or asymptomatic for the development of collateral circulation could reestablish portal flow to the liver” Consider - Patients with late phase PVT after LT may present with liver dysfunction, portal hypertension, or they may be asymptomatic due to the development of collateral circulation that could reestablish portal flow to the liver.
- For the figure: consider using the term asymptomatic instead of “asymptom” and during what time frame is this post-transplant?
- You use the terminology ‘Yerdel stage’ – at least include a citation for this staging system and consider explaining this system in the manuscript
- Table 2: please check spelling and spacing – eg patient is spelled incorrectly for Jannet et al
- Table 3: consider reporting male gender with n (%) to make this more clear
